# Comparison of GFAP and UCH-L1 Measurements Using Two Automated Immunoassays (i-STAT^®^ and Alinity^®^) for the Management of Patients with Mild Traumatic Brain Injury: Preliminary Results from a French Single-Center Approach

**DOI:** 10.3390/ijms25084539

**Published:** 2024-04-21

**Authors:** Charlotte Oris, Clara Khatib-Chahidi, Bruno Pereira, Valentin Bailly Defrance, Damien Bouvier, Vincent Sapin

**Affiliations:** 1Biochemistry and Molecular Genetics Department, University Hospital, 63000 Clermont-Ferrand, France; coris@chu-clermontferrand.fr (C.O.); ckhatibchahidi@chu-clermontferrand.fr (C.K.-C.); valentin.bailly-defrance@etu.uca.fr (V.B.D.); dbouvier@chu-clermontferrand.fr (D.B.); 2Clinical Research and Innovation Department, University Hospital, 63000 Clermont-Ferrand, France; bpereira@chu-clermontferrand.fr; 3Institute “Genetic, Reproduction and Development”, UMR INSERM 1103 CNRS 6293, Université Clermont Auvergne, 63001 Clermont-Ferrand, France

**Keywords:** ubiquitin carboxy-terminal hydrolase L1, glial fibrillary acidic protein, mild traumatic brain injury, biological diagnosis

## Abstract

The measurement of blood glial fibrillary acidic protein (GFAP) and ubiquitin carboxy-terminal hydrolase L1 (UCH-L1) may assist in the management of mild traumatic brain injury (mTBI). This study aims to compare GFAP and UCH-L1 values measured using a handheld device with those measured using a core laboratory platform. We enrolled 230 mTBI patients at intermediate risk of complications. Following French guidelines, a negative S100B value permits the patient to be discharged without a computed tomography scan. Plasma GFAP and UCH-L1 levels were retrospectively measured using i-STAT^®^ and Alinity^®^ i analyzers in patients managed within 12 h post-trauma. Our analysis indicates a strong correlation of biomarker measurements between the two analyzers. Cohen’s kappa coefficients and Lin’s concordance coefficients were both ≥0.7, while Spearman’s correlation coefficient was 0.94 for GFAP and 0.90 for UCH-L1. Additionally, the diagnostic performance in identifying an intracranial lesion was not significantly different between the two analyzers, with a sensitivity of 100% and specificity of approximately 30%. GFAP and UCH-L1 levels measured using Abbott’s i-STAT^®^ and Alinity^®^ i platform assays are highly correlated both analytically and clinically in a cohort of 230 patients managed for mTBI according to French guidelines.

## 1. Introduction

Mild traumatic brain injury (mTBI) constitutes about 80% of all TBIs and is an increasingly common cause of morbidity and mortality worldwide [1]. The gold standard for diagnosing intracranial injuries such as hemorrhage or edema in patients with mTBI is cranial computed tomography (CT), which detects such injuries in 6 to 10% of all CT scans performed [2]. However, CT imaging must be used appropriately to avoid preventable adverse health outcomes. It is a costly, labor-intensive, and time-consuming medical test that can lead to overcrowding in emergency departments and medical imaging departments [3,4]. Furthermore, various recent epidemiological studies have reported a correlation between radiation exposure from CT scans in children and the risk of developing cancer in the future [5].

In this context, it has become clear that new diagnostic tools need to be incorporated into the management of mTBI. In recent years, research has focused mainly on blood biomarkers. In Europe, guidelines [6,7] recommend the measurement of S100B protein, which has led to a 30% reduction in the use of CT scans [8,9,10]. However, this biomarker has been criticized for its limited neurospecificity and rapid half-life, requiring blood sampling within 3 h of trauma [8,11]. In 2018, the US Food and Drug Administration approved GFAP and UCH-L1 assays as an aid in assessing mTBI [12,13,14]. The ALERT-TBI study demonstrated that blood tests, including GFAP and UCH-L1, outperformed those of S100B in terms of sensitivity and specificity (40%) [15]. To date, only two published studies have examined the performance of the GFAP and UCH-L1 blood tests using an automated assay (i-STAT^®^, Abbott) in a routine clinical setting [15,16]. Compared to S100B, GFAP and UCH-L1 can be measured within 12 h of head injury, making them suitable for a larger number of patients (20% in our experience) and reducing the need for CT scans.

Abbott has recently introduced two automated immunoassays for the routine determination of GFAP and UCH-L1 levels. One is a portable i-STAT^®^ point-of-care device, and the other is an Alinity^®^ i central laboratory platform. The two systems are complementary and their use would democratize biomarker measurement and optimize patient management. However, it is currently unknown whether GFAP and UCH-L1 levels measured on these different systems are commutable. The objective of this study is to compare the GFAP and UCH-L1 values obtained from the i-STAT^®^ and Alinity^®^ analyzers for the first time and to assess their level of agreement both analytically and clinically.

## 2. Results

### 2.1. Patient Characteristics

Between January 2023 and June 2023, 230 patients admitted to the emergency department for mTBI met the inclusion criteria and were enrolled in the study. The group included 104 females (45.2%) and 126 males (54.8%), resulting in a sex ratio of 1.2. The median age of the participants was 66.2 years (minimum: 18.2; maximum: 101.2; IQR: 33.9–82.8). On arrival at the emergency department, 220 patients (95.7%) had a GCS score of 15, 9 patients (4.0%) had a score of 14, and only 1 patient (0.3%) had a score of 13. The blood test was conducted at a median interval of 101 min after the trauma (min: 25, max: 720, IQR: 75–162). Out of the 230 patients with mTBI, 219 (95%) were classified as ICL− (intracranial lesion) and 11 (5%) as ICL+. The latter group had the following lesions: five subdural hematomas (SDH) and six subarachnoid hemorrhages (SAH). Of the total number of patients, 194 (84.3%) were discharged, 35 (15.3%) remained in hospital for observation, and 1 patient died (0.4%) (Table 1).

### 2.2. Analytical Correlation and Concordance

#### 2.2.1. GFAP

There was no significant difference in the median GFAP levels measured by the two analyzers, i-STAT^®^ and Alinity^®^ i. The levels were 43 ng/L (min: 30; max: 907; IQR: 30–76) and 46 ng/L (min: 1; max: 961; IQR: 23–85), respectively (*p* = 0.17) (Table 2). The Cohen’s kappa coefficients and Lin’s concordance coefficients showed a strong agreement between the two analyzers, with values largely superior to 0.7. Additionally, the agreement rate between the analyzers was high, at 94% (Table 2). The Bland–Altman plot shows a mean difference of −1.84 ng/L, with 95% limits of agreement ranging from −71 to 67 ng/L (Figure 1). The two assays were strongly correlated with a slope of 0.96, intercept of 4.44, and a Spearman’s correlation coefficient (ρ) of 0.94 (Figure 2).

#### 2.2.2. UCH-L1

Median UCH-L1 levels measured by the two analyzers, i-STAT^®^ and Alinity^®^, were not significantly different. The levels were 282 ng/L (min: 200; max: 2376; IQR: 200–472) and 350 ng/L (min: 42; max: 2940; IQR: 200–548), respectively (*p* = 0.14) (Table 2). The Cohen’s kappa coefficients and Lin’s concordance coefficients were both ≥0.7, indicating good agreement between the two analyzers. Additionally, the agreement rate between the analyzers was high at 86% (Table 2). The Bland–Altman plot shows a mean difference of −32 ng/L, with 95% limits of agreement ranging from −366 to 301 ng/L (Figure 1). The two assays were strongly correlated, with a slope of 0.99, intercept of 32.41, and a Spearman’s correlation coefficient (ρ) of 0.90 (Figure 2).

### 2.3. Biomarker Levels According to Imaging Findings on CT-Scan

#### 2.3.1. i-STAT^®^ Analyzer

The median GFAP concentration in the ICL− group (41 ng/L, min: 30, max: 507, IQR: 30–74) was significantly lower than that in the ICL+ group (85 ng/L, min: 30, max: 907, IQR: 59–179) (Table 3) (*p* = 0.003). There was no significant difference in the median UCH-L1 values between the ICL− and ICL+ groups (279 ng/L, min: 200, max: 2376, IQR: 200–471 vs. 312 ng/L, min: 200, max: 1931, IQR: 281–540) (*p* = 0.20) (Table 3).

#### 2.3.2. Alinity^®^ i Analyzer

The GFAP concentration median in the ICL− group significantly differed from that in the ICL+ group 44 ng/L (min: 1, max: 546, IQR: 23–83) vs. 87 ng/L (min: 29, max: 961, IQR: 71–165) (*p* = 0.005) (Table 3). The median UCH-L1 values in the ICL− and ICL+ groups did not significantly differ 342 ng/L (min: 42, max: 2940, IQR: 197–548 vs. 469 ng/L (min: 187, max: 1603, IQR: 282–570) (*p* = 0.14) (Table 3).

### 2.4. Clinical Concordance for the Combined Test “GFAP + UCH-L1”

The combination of GFAP and UCH-L1 measurements accurately identified ICL+ patients using both the i-STAT^®^ and the Alinity^®^ i analyzers. The i-STAT^®^ analyzer had a sensitivity of 100%, a specificity of 28.8%, and a negative predictive value of 100%. The Alinity^®^ i analyzer had a sensitivity of 100%, a specificity of 29.7%, and a negative predictive value of 100%. There was no significant difference in diagnostic performance between the two systems (*p* = 0.68; *p* = 1) as shown in Table 4. The Cohen’s kappa coefficient value of 0.74 and the high percentage of identical interpretations (90%) indicate good agreement between the two analyzers.

## 3. Discussion

To manage patients suffering from mild traumatic brain injury, Abbott has recently validated automated GFAP and UCH-L1 immunoassays for routine clinical use. Two systems are available: the i-STAT^®^ handheld device for point-of-care and the Alinity^®^ i platform for core lab use. For the first time, we compared two different immunoassays to determine their degree of agreement both analytically and clinically. We used a cohort of 230 mTBI patients managed according to French Society of Emergency Medicine guidelines [7]. Our analyses demonstrate a strong analytical correlation of biomarker measurements between the two analyzers. Although the methods are correlated, they require the use of reference values adapted to each assay. Validated cutoff values for iSTAT ^®^ are 30 ng/L and 360 ng/L for GFAP and UCH-L1, respectively, compared to 35 ng/L and 400 ng/L for Alinity^®^ i. When these thresholds were applied in our cohort, the diagnostic performance was found to be identical between the two analyzers. This demonstrates the high clinical concordance of the two systems, with a sensitivity and negative predictive value of 100% and a specificity of approximately 30%.

To date, only two studies have been published on the diagnostic performance of the GFAP and UCH-L1 blood tests using an automated assay (i-STAT^®^, Abbott) in a routine clinical setting [15,16]. In the first study, the combined measurement of GFAP and UCH-L1 outperformed S100B with equivalent sensitivity and higher specificity (40%) [15]. In our cohort, the diagnostic performance of GFAP and UCH-L1 were comparable to that of S100B protein [16]. Numerous observational studies, several interventional studies, and a meta-analysis have reported a sensitivity of 100% for S100B protein with a specificity of approximately 30% [8,10]. However, unlike S100B, GFAP and UCH-L1 have a longer half-life [17,18]. Therefore, according to the French guidelines, a combined measurement of GFAP and UCH-L1 is recommended for patients managed within 12 h of mTBI, as opposed to 3 h for S100B [7]. This approach increases the number of patients who can benefit from the measurement of GFAP and UCH-L1 levels by 20% compared to S100B, and optimizes the reduction in CT scans. Biberthaler et al. [19] have recently highlighted the clinical utility of GFAP and UCH-L1 as biomarkers for ruling out CT-positive injury in acute mTBI. However, in their cohort of 109 patients, they only presented the AUCs of the biomarkers without mentioning their diagnostic performance for clinical application. It is important to emphasize the value of using these biomarkers in combination, as recommended by the FDA and French guidelines. In our study, we found that UCH-L1 was not able to differentiate between ICL+ patients and other patients (ICL−). However, as mentioned previously, the combination of the two biomarkers has suitable diagnostic performance for routine use. Current strategies aim to increase the specificity of brain damage biomarkers by designing new decisional cut-offs linked to the age of patients (a parameter well described to increase normal values) and by combining their use with inflammatory biomarkers such as blood interleukin-10 concentrations [20,21,22].

This study evaluates the performance of GFAP and UCH-L1 levels measured with the Alinity^®^ system in patients with mTBI. In 2021, Korley et al. compared GFAP and UCH-L1 values measured using the point-of-care i-STAT^®^ handheld device and the core lab Architect^®^ platform [23]. The authors reported a strong correlation between GFAP and UCH-L1 values measured by the two analyzers and suggested that values from one system could be converted to the other. The Architect^®^ platform is no longer marketed by Abbott, but it is still available in some routine laboratories. The Alinity^®^ is the new multimodule system currently available on the market, providing enhanced productivity for laboratories with high-volume immunoassay demands. The i-STAT^®^ is valuable for its complementarity with the Alinity^®^. Currently, the i-STAT^®^ requires an EDTA plasma sample. However, Abbott is developing a test that uses a whole blood sample and announced on 1 April 2024 the clearance of such strategy by the FDA. This advancement would allow the device to be used at the point of care in healthcare settings. The use of this system in the ED could optimize the management of patients with mTBI. Together, this information could support the management of patients with a combination of handheld devices for point-of-care and core lab systems. This statement could be useful in creating a network of coherent clinical and biological approaches in a given territory, leading to a homogeneous management of patients with mTBI. The study of medical optimization also had to consider cost-saving measures. A recent study demonstrated the medico-economic benefits of the biomarker strategy compared to the classic CT-scan approach [24]. Additionally, this could serve as a valuable biological tool and strategy to investigate the benefits of determining GFAP and UCH-L1 levels in the blood for other neurological conditions such as severe TBI [25], stroke [26], and intracranial pressure [27]. Our study has limitations. Firstly, the results of this analysis only apply to values obtained from the i-STAT^®^ handheld device and the Alinity^®^ system for the dual determination of GFAP and UCH-L1. Secondly, i-STAT^®^ values were generated using plasma samples. It is unknown whether the correlation between i-STAT^®^ and Alinity^®^ values will hold when whole blood is used for i-STAT^®^ measurements. One limitation of our study is the relatively small sample size of patients compared to studies with larger cohorts. However, our findings are consistent with previously published data for mTBI [2]. In our study, 5% of patients had positive CT scans, which provides external validation of our cohort.

## 4. Materials and Methods

### 4.1. Study Design and Patients

The study was carried out in the Emergency Department (ED) of the University Hospital of Clermont-Ferrand in France from January 2023 to June 2023. It was approved by the CPP Ile-de-France X (reference 52-2019) and adhered to the ethical principles for medical research outlined in the Declaration of Helsinki. Subjects were informed of their right to refuse the use of their clinical data for research purpose. The study included adult patients (18 years old and older) with mild traumatic brain injury (mTBI) who were classified as moderate risk for intracranial injury based on the criteria established by the French Society of Emergency Medicine (SFMU) [7]. The patients included in the study had a Glasgow Coma Scale (GCS) score ranging from 13 to 15 and presented with at least one associated risk factor, such as antiplatelet monotherapy, loss of consciousness, or post-traumatic amnesia of facts 30 min before the injury. Following the validated French guidelines by the SFMU, we performed routine S100B measurement on the automated Cobas^®^ system (Roche Diagnostics, Meylan, France) for patients who underwent blood sampling within 3 h of mTBI. Patients with S100B levels above the decision threshold of 0.10 µg/L underwent CT scanning. People with S100B levels below the cut-off may be considered at low risk of intracranial complications and released. All patients underwent venipuncture for blood biomarker measurement 12 h after mTBI for GFAP and UCH-L1. GFAP and UCH-L1 samples were stored at −80 °C until analysis, whereas S100B assays were performed within 2 h of collection for routine clinical use. The clinical management of the patients was not affected by the GFAP and UCH-L1 results.

### 4.2. GFAP and UCH-L1 Assays

#### 4.2.1. i-STAT^®^ Analyzer

Plasma concentrations of GFAP and UCH-L1 were measured using i-STAT^®^ TBI Plasma cartridges (Abbott, Abbot Park, IL, USA) on the i-STAT^®^ analyzer. The manufacturer’s instructions were followed, and the i-STAT^®^ was calibrated using the manufacturer’s standards. The assay required 20 µL of plasma and lasted 15 min. The lower limit of quantification for GFAP was 23.0 ng/L, and the upper limit was 10,000 ng/L. The lower limit of quantification for UCH-L1 was 70.0 ng/mL, and the upper limit was 3200 ng/mL. The CV of the precision within the assay was typically less than 10%. The normal values for GFAP and UCHL1 were set at less than 30 ng/mL and 360 ng/mL, respectively, in accordance with the recommendations of Abbott.

#### 4.2.2. Alinity^®^ Analyzer

Plasma concentrations of GFAP and UCH-L1 were measured using the Alinity^®^ analyzer, following the manufacturer’s instructions. The analyzer was calibrated with the manufacturer’s standards. The assay required 300 μL of plasma and lasted 18 min. GFAP had a lower limit of quantitation of 6.1 ng/mL and an upper limit of 42,000 ng/mL. UCH-L1 had a minimum and maximum quantifiable value at 26.3 ng/L and 25,000 ng/L respectively. The typical within-assay precision CV was less than 5%. According to Abbott recommendations, normal levels for GFAP and UCHL1 were set at less than 35 ng/L and 400 ng/L, respectively.

### 4.3. Cranial Computed Tomography Scan

The CT scan was conducted using a GE Healthcare Revolution GSI^®^ (Chicago, IL, USA) with the following protocol: helical mode, 2.25 mm slice thickness, 1.25 mm interval, 120 kV, and a maximum of 280 mA from C1 to the top of the head. Additional bone window reconstructions were also performed. To determine whether patients had a trauma-related intracranial lesion, we collected radiological parameters and divided them into two groups: those with no evidence of trauma-related intracranial lesions on CT (ICL−) and those with at least one trauma-related intracranial lesion on CT (ICL+). A CT scan was considered to be positive if there was any evidence of intracranial pathology, such as a hematoma, air, or contusion. Furthermore, patients with an S100B concentration below the decision threshold of 0.10 µg/L who did not undergo a CT scan were classified as ICL due to the high sensitivity of S100B. It is important to note that this classification is objective and based solely on the S100B concentration.

### 4.4. Statistical Analysis

The statistical distribution of continuous data was expressed using median, minimum (min), maximum (max), and interquartile range (IQR). The normality assumption was assessed using the Shapiro–Wilk test. The agreement between two methods was evaluated using Pearson’s correlation coefficient and Lin’s concordance correlation coefficient for UCHL1 and GFAP treated as continuous variables. For UCHL1 and GFAP treated as categorical variables (dichotomized according to thresholds), the agreement was evaluated using agreement rate (%) and Kappa concordance coefficient. The results were interpreted according to the recommendations from Altman and colleagues [28]: <0.4 indicates no agreement, [0.4–0.7] indicates poor agreement, and ≥0.7 indicates good to strong agreement. Additionally, for each biomarker, a comparison of two methods was performed using linear regression, Bland–Altman plot, and paired comparisons conducted with the Wilcoxon test. The results were expressed using the difference between methods and the 95% limits of agreement interval. To compare UCHL1 and GFAP values according to the gold standard, the Mann–Whitney test was used, as the assumptions to apply the Student’s *t*-test were not met. The study estimated sensitivity (SE), specificity (SP), positive predictive value (PPV), and negative predictive value (NPV) with 95% confidence intervals. The McNemar test and a generalized linear mixed model with a logit link function were used to compare the two methods for the aforementioned diagnostic values. Statistical analyses were conducted using Stata software (version 15, StataCorp, College Station, TX, USA). All statistical tests were performed with a two-sided type I error rate of 5%.

## 5. Conclusions

We demonstrated the analytical correlation between the i-STAT^®^ handheld device and the Alinity^®^ central laboratory platform for the plasma measurement of GFAP and UCH-L1. However, this correlation alone is not enough to determine that the clinical decision will be the same with either analyzer. We compared the clinical performance of the two systems and found them to be identical. In a cohort of 230 mTBI patients managed according to SFMU guidelines, both systems had a sensitivity and negative predictive value of 100%, while the specificity was approximately 30%. Finally, our study indicates that it may be feasible to use either the Alinity^®^ analyzer or the i-STAT^®^ handheld device in a routine laboratory for the simultaneous measurement of GFAP and UCH-L1 in the management of patients with mTBI. This could also lead to a more standardized approach to the clinical and biological aspects of mTBI management in areas with varying degrees of analyzer availability.

## Figures and Tables

**Figure 1 ijms-25-04539-f001:**
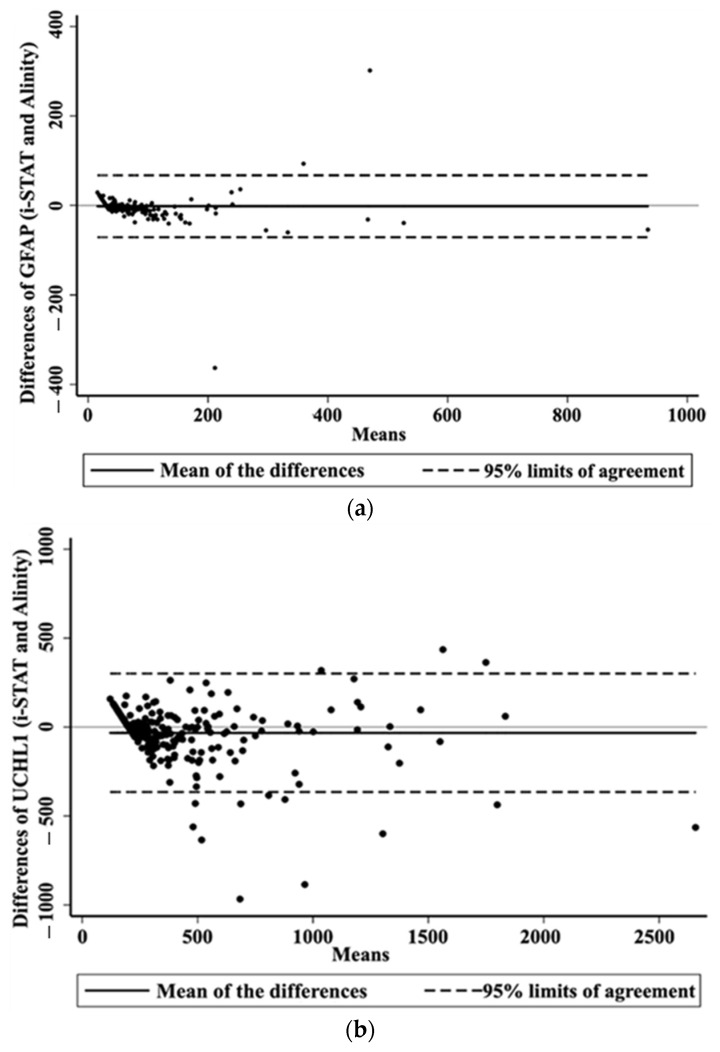
Bland–Altman plots of GFAP (**a**) and UCH-L1 (**b**) values measured by the i-STAT^®^ and Alinity^®^ i assays.

**Figure 2 ijms-25-04539-f002:**
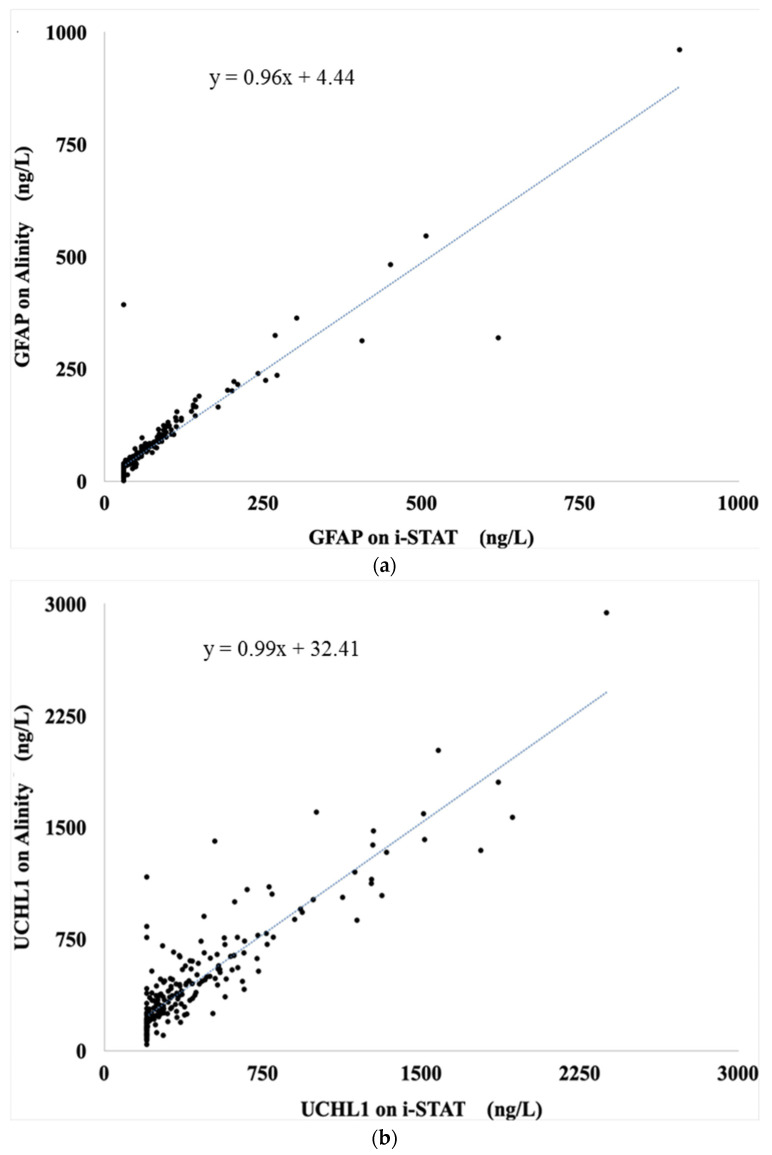
Linear regressions of GFAP (**a**) and UCH-L1 (**b**) values measured by the i-STAT^®^ and Alinity^®^ i assays.

**Table 1 ijms-25-04539-t001:** Demographic characteristics, clinically relevant information, and radiological findings for the whole study population.

Demographic Characteristics and Clinically Relevant Information	
Total	230
Sex ratio (M/F)	1.2
Mean age in years (min; max; IQR)	66.2 (18.2; 101.2; 33.9–82.8)
Mean sampling time in minutes (min; max; IQR)	100.5 (25; 720; 75–162)
GCS at admission, *n* (%)	
13	1 (0.3)
14	9 (4)
15	220 (95.7)
Clinical outcome, *n* (%)	
Discharged	194 (84.3)
Hospitalized for surveillance	35 (15.3)
Death	1 (0.4)
CT+ patients, *n* (%)	
Subdural hematoma	5 (2.2)
Subarachnoid hemorrhage	6 (2.6)

CT, computed tomography; GCS, Glasgow Coma Scale; IQR, interquartile range; max, maximum; min, minimum.

**Table 2 ijms-25-04539-t002:** Analytical correlation and concordance between i-STAT^®^ and Alinity i^®^.

	GFAP	UCH-L1
i-STAT^®^ (ng/L), median	43	282
(min; max; IQR)	(30; 907; 30–76)	(200; 2376; 200–472)
Alinity^®^ i (ng/L), median	46.3	350
(min; max; IQR)	(1; 961; 23–85)	(42; 2940; 200–548)
*p*-Value	0.17	0.14
Lin (95% CI)	0.93 (0.91–0.95)	0.89 (0.87–0.92)
% of agreement	94	86
Kappa (95% CI)	0.88 (0.82–0.94)	0.70 (0.60–0.79)

IQR, interquartile range; max, maximum; min, minimum; CI, confidence interval; a *p*-value ≤ 0.05 is considered statistically significant.

**Table 3 ijms-25-04539-t003:** Biomarker levels according to imaging findings on CT-scan.

**i-STAT^®^**			
	**ICL− (*n =* 219)**	**ICL+ (*n =* 11)**	***p*-Value**
GFAP in ng/L median (min; max; IQR)	41 (30; 507; 30–74)	85 (30; 907; 59–179)	0.003
UCH-L1 in ng/L median (min; max; IQR)	279 (200; 2376; 200–471)	312 (200; 1931; 281–540)	0.20
**Alinity^®^ i**			
	**ICL− (*n =* 219)**	**ICL+ (*n =* 11)**	***p*-Value**
GFAP in ng/L median (min; max; IQR)	44 (1;546; 23–83)	87 (29; 961; 71.1–165)	0.005
UCH-L1 in ng/L median (min; max; IQR)	342 (42; 2940; 197–548)	469 (187; 1603; 282–570)	0.14

ICL−, patients without any sign of trauma-relevant intracranial lesions on the CT scan; ICL+, patients with at least one pathophysiological trauma-relevant lesion intracranial found on the CT scan; IQR, interquartile range; max, maximum; min, minimum; a *p*-value ≤ 0.05 is considered statistically significant.

**Table 4 ijms-25-04539-t004:** Diagnostic performance for the combined test “GFAP + UCH-L1”.

	GFAP + UCH-L1 (i-STAT^®^)	GFAP + UCH-L1 (Alinity^®^ i)	*p*-Value
SE (95% CI)	100% (75.1–100%)	100% (72–100%)	1
SP (95% CI)	28.8% (22.9–35.3%)	29.7% (23.7–36.2%)	0.68
PPV (95% CI)	6.6% (3.3–11.5%)	6.7% (3.3–11.5%)	0.98
NPV (95% CI)	100% (94.3–100%)	100% (94.5–100%)	1

CI, confidence interval; NPV, negative predictive value; PPV, positive predictive value; SE, sensitivity; SP, specificity; a *p*-value ≤ 0.05 is considered statistically significant.

## Data Availability

Data are contained within this article.

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
