# Peer review of "Comparison of GFAP and UCH-L1 Measurements Using Two Automated Immunoassays (i-STAT® and Alinity®) for the Management of Patients with Mild Traumatic Brain Injury: Preliminary Results from a French Single-Center Approach"

_ijms, 2024, doi:10.3390/ijms25084539_

Round 1
Reviewer 1 Report
Comments and Suggestions for Authors
In the present manuscript, the authors compare blood glial fibrillary acidic protein (GFAP) and ubiquinonin carboxy-terminal hydrolase L1 (UCH-L1) values using handheld devices and core laboratory platforms. 230 patients at intermediate risk of complications were enrolled. Plasma GFAP and UCH-L1 levels were measured using i-STAT® and Alinity® i analyzers. Results showed strong correlation between the two analyzers, with a sensitivity of 100% and specificity of approximately 30%.
The article is well written and clear.
Author Response
The authors express their gratitude to the reviewer for their constructive evaluation of the manuscript.
Reviewer 2 Report
Comments and Suggestions for Authors
The paper entitled "Comparison of GFAP and UCH-L1 measurements using two automated immunoassays (i-STAT® and Alinity®) for the management of patients with mild traumatic brain injury" is interesting. The matter is very relevant in public health.
The manuscript's structure is well-organized, the data is well-explained, and the statistical analysis is coherent with the results. The introduction and discussion are supported by literature. The paper's results impact the outcome of patients with mild TBI.
Congratulations to the authors for the valuable paper. I have no comments to make, I recommend accepting the paper.
Comments on the Quality of English Language
Minor editing of English language required
Author Response
The authors express their gratitude to the reviewer for their positive evaluation and review of the manuscript. A minor revision of the English language was conducted.
Reviewer 3 Report
Comments and Suggestions for Authors
Reviewing the provided paper titled "Comparison of GFAP and UCH-L1 measurements using two automated immunoassays (i-STAT® and Alinity®) for the management of patients with mild traumatic brain injury" presents a detailed analysis of the potential for GFAP and UCH-L1 blood biomarkers in enhancing diagnostic protocols for mild traumatic brain injury (mTBI). The authors have aimed to investigate the analytical and clinical correlation between measurements obtained from two different automated immunoassay platforms. Here are some major comments and suggestions for the authors:
Strengths:
1. Significant Contribution to mTBI Management: The study addresses a critical area in emergency medicine by exploring less invasive, rapid diagnostic alternatives to computed tomography (CT) scans for mTBI patients. The use of GFAP and UCH-L1 as biomarkers could significantly impact patient management protocols, potentially reducing unnecessary radiation exposure and resource utilization in emergency departments.
2. Robust Methodological Framework: The methodology, including patient selection, biomarker measurement, and statistical analysis, is well-detailed and appropriate. The comparison of i-STAT® and Alinity® platforms through various statistical measures (Cohen's kappa, Lin's concordance, Spearman's correlation) provides a comprehensive assessment of their agreement and diagnostic performance.
3. Clear Presentation of Results: The results are presented in a clear, logical manner, with effective use of tables and figures to summarize key findings. The demonstration of high analytical correlation and clinical concordance between the two platforms is a crucial finding that supports the potential interchangeable use of these assays in clinical practice.
Major concern: "sensitivity of 100% and specificity 25
of approximately 30%". First, report and visualize some more details about this result. how and why it is possible? Second, this aspect warrants a more thorough discussion, particularly regarding the implications for false positives and the potential burden on healthcare resources. Pleasw also discuss strategies to improve specificity or combine these biomarkers with other diagnostic criteria.
2. Exploration of Clinical Utility Beyond mTBI: The authors focus on the application of GFAP and UCH-L1 in mTBI. However, a discussion on the potential utility of these biomarkers in other neurological conditions (e.g., severe TBI, stroke) could broaden the relevance of the findings. Additionally, considering the evolving landscape of TBI management, future research directions or potential technological advancements in biomarker detection could be explored. Because other studies also highlighted the strong biomarker role of intracranial pressure and intracranial compliance for TBI, raised ICP, hydrocephalus, and some brain disorders [10.1038/s42003-022-04128-8]. You can discuss ir more.
3. Limitations Related to the Study Population and Setting: The study is conducted within a single institution, and the sample size, while substantial, may not fully represent the broader population of mTBI patients. Discussing the potential impact of demographic variability (age, sex, comorbidities) on biomarker levels and diagnostic performance could provide insights into the generalizability of the findings.
4. Consideration of Economic and Operational Factors: Implementing new diagnostic assays in clinical practice involves considerations beyond analytical and clinical performance, including cost, operational logistics, and training requirements. A discussion on these practical aspects, possibly including a cost-benefit analysis or considerations for integration into existing clinical workflows, would enhance the paper's relevance to healthcare decision-makers.
5. Addressing the Evolution of mTBI Guidelines: The study references current French guidelines for mTBI management. Given the dynamic nature of clinical guideline development, acknowledging ongoing or future revisions to these guidelines in response to emerging evidence (including studies like this one) could contextualize the study within the larger continuum of TBI care evolution.
Comments on the Quality of English LanguageMedium level
Author Response
The authors express their gratitude to the reviewer for identifying the strengths of their manuscript.
Regarding the sensitivity of 100% and specificity of approximately 30%, the clinical performance was consistent across all studies involving S100B and mTBI. However, it should be noted that S100B lacks neuro-specificity due to its expression in other tissues or organs, as reported in Protein Atlas. The most crucial clinical performance is sensitivity, which is essential for avoiding false negative detection of brain lesions after mTBI. The biomarker must have a negative predictive value of 100% to be effective. With a specificity of around 30%, it can already eliminate approximately 30% of unnecessary CT scans for ruling out patients.
Two strategies are currently being explored to improve specificity: the use of specific decisional cut-offs based on patient age and the addition of complementary blood biomarkers, such as neuro-inflammatory biomarkers (e.g. interleukin 10), which can increase specificity to 40-50%. As requested by the reviewer, this point has been included in the revised version (lines 227-231).
The authors concur with the reviewer's suggestion that the applications of GFAP and UCHL1 could be extended. Although our manuscript focuses on mTBI, we also mention other potential indications for blood determinations of GFAP and UCH-L1 for other neurological pathologies, such as severe TBI, stroke, or intracranial pressure. These pathologies have already been shown to be optimized by measuring such blood biomarkers (lines 252 to 255). Since the submission of our manuscript, an interesting technological advancement has been made. The FDA has announced the clearance for a whole blood rapid test, which can be used at the patient's bedside to assess concussion without the need for centrifugation. This technological step could enable the use of real POCT in clinical departments. We are implementing an important evolution of the test in this revised version (line 243).
One of the main advantages of using S100B, GFAP, and UCH-L1 biomarkers is their robustness in clinical settings, regardless of the size of the cohort. They exhibit similar levels of specificity, sensitivity, negative predictive value, and positive predictive value with 100 patients as with 1500 patients. Our cohort was qualified based on the percentage of intracranial lesions reported in other studies (approximately 5 to 10%), and our clinical performances fall within the published ranges. Regarding demographic variability, all previous studies found no influence of sex or other comorbidities, except for neurodegenerative pathologies which were excluded in the cohort used in this study. The only parameter that was found to physiologically modify the concentrations of the three biomarkers was age, as mentioned in our response to another comment (lines 229-230). In response to another point raised by the reviewer, we have added 'monocentric' to the title to clarify the study's location.
The authors acknowledge the reviewer's comment regarding the cost-benefit analysis of implementing blood biomarkers for managing mTBI patients. While such studies are rare, it would be valuable to highlight the medico-economic benefits of this biomarker strategy compared to the classic CT-scan approach, as suggested by the reviewer (lines 249-252).
The French professional recommendations for managing patients with mTBI are recent, having been released at the end of 2022. The recommendations mention the well-established S100B biomarker and the emerging biomarkers GFAP and UCH-L1. Based on our analysis, we do not anticipate any modifications to these recommendations in the near future, so it is not necessary to indicate such an evolution. The corresponding author of this manuscript is the coordinator of the Working Group 'mTBI Biomarkers' of the European Federation of Laboratory Medicine. This publication presents the French experience, which will be shared with other countries. However, it is too early to suggest any modifications to the larger European continuum at this stage.
A new edition of the English language has been produced.
Reviewer 4 Report
Comments and Suggestions for Authors
This is a very well written manuscript, which aims to elucidate the potential role of GFAP and UCH-L1 measurements in blood serum, in patients suffering from mild traumatic brain injury. The main target of this study is to specify the existence of any potential role of such measurements in the management of mild TBI, for patients managed within 12 hours of mTBI, as opposed to 3 hours for S100B, as is already known. The data analysis is adequate and the conclusions are adequately supported. Nevertheless, I would like to mention that there are some recommendations from my side regarding some methodological errors that are encountered.
According to the authors, the size of the patient's sample is relatively restricted. Because of that, the title of the paper should be modified, including a statetement such as' preleminary results from a single center study'.
According to my point of view, the 'Materials and Methods' section should not follow the 'Discussion' section. Instead of that, the opposite seems to be more acceptable and it is highly recommended.
Comments on the Quality of English LanguageAccording to my opinion, only a minor editing of English language is required.
Author Response
The authors express their gratitude to the reviewer for their constructive evaluation of the manuscript. The title was revised to incorporate the concepts of preliminary findings and a single-centre approach, as suggested by the reviewer.
The authors were unable to modify this aspect as it is the form specified in the author instructions.
A new edition of the English language has been produced.